# *POWR1* is a domestication gene pleiotropically regulating seed quality and yield in soybean

Wolfgang Goettel[1,9], Hengyou Zhang [2,3,9], Ying Li[2], Zhenzhen Qiao[2], He Jiang[2], Dianyun Hou[1,4], Qijian Song[5], Vincent R. Pantalone[6], Bao-Hua Song [7], Deyue Yu [8] & Yong-qiang Charles An [1,2✉]

Seed protein, oil content and yield are highly correlated agronomically important traits that essentially account for the economic value of soybean. The underlying molecular mechanisms and selection of these correlated seed traits during soybean domestication are, however, less known. Here, we demonstrate that a CCT gene, *POWR1*, underlies a large-effect protein/oil QTL. A causative TE insertion truncates its CCT domain and substantially increases seed oil content, weight, and yield while decreasing protein content. *POWR1* pleiotropically controls these traits likely through regulating seed nutrient transport and lipid metabolism genes. *POWR1* is also a domestication gene. We hypothesize that the TE insertion allele is exclusively fixed in cultivated soybean due to selection for larger seeds during domestication, which significantly contributes to shaping soybean with increased yield/seed weight/oil but reduced protein content. This study provides insights into soybean domestication and is significant in improving seed quality and yield in soybean and other crop species.

[1] US Department of Agriculture, Agricultural Research Service, Midwest Area, Plant Genetics Research Unit, 975N Warson Rd, St. Louis, MO 63132, USA. [2] Donald Danforth Plant Science Center, 975N Warson Rd, St. Louis, MO 63132, USA. [3] Key Laboratory of Soybean Molecular Design Breeding, Northeast Institute of Geography and Agroecology, Chinese Academy of Sciences, Harbin 150081, China. [4] College of Agriculture, Henan University of Science and Technology, Luoyang, Henan 471023, China. [5] US Department of Agriculture, Agricultural Research Service, Soybean Genomics and Improvement Laboratory, Beltsville, MD 20705, USA. [6] Department of Plant Sciences, University of Tennessee, Knoxville, TN 37996, USA. [7] Department of Biological Sciences, University of North Carolina at Charlotte, Charlotte, NC 28223, USA. [8] National Center for Soybean Improvement, National Key Laboratory of Crop Genetics and Germplasm Enhancement, Nanjing Agricultural University, Nanjing, Jiangsu 210095, China. [9] These authors contributed equally: Wolfgang Goettel, Hengyou Zhang. ✉email: yong-qiang.an@usda.gov

Soybean [*Glycine max* (L.) Merr.] is one of the most important seed crops grown worldwide. It was domesticated from wild soybean (*Glycine soja* Sieb. & Zucc.) in East Asia about 6000–9000 years ago. Domestication and improvement have shaped soybean as the most important dual-function crop to provide highly valuable seed protein and oil, which together account for its high economic value[1,2].

Seed protein content, oil content, and yield are considered as the most important traits in soybean. The commodity-type soybean varieties typically contain ~40% seed protein and 20% seed oil. Cultivated soybeans often contain higher seed yield and oil content, but lower protein content than their ancestral wild soybeans[3]. However, these three traits vary greatly among soybean accessions. The seed traits are often correlated with each other, where seed protein content frequently shows a negative correlation with both seed oil content and yield[4–10]. The inverse relationship poses a great challenge to simultaneously improve the seed quality traits and yield to enhance the overall economic value of soybean. Illustrating the genetic and molecular basis underlying the three interrelated traits and understanding how those interrelated and agronomically important traits have been selected over the course of soybean domestication and improvement therefore holds great significance for seed quality and yield improvement in soybean.

In this study, we demonstrate that a CCT-domain gene, *POWR1* (*Protein, Oil, Weight, Regulator 1*), underlies a large-effect protein and oil QTL on chr20. A TE (transposable element) insertion in the conserved CCT domain leads to increased seed oil content, seed weight, and decreased seed protein content. *POWR1* is preferentially expressed in developing seed coat and controls the seed traits likely through regulating nutrient transport and lipid metabolism. *POWR1* is a domestication gene, and the mutated allele is completely fixed in cultivated soybean except that a few Asian cultivated soybean accessions contain the wild-type high-protein *POWR1* allele from post-domestication introgression. The reversal of the domestication process and our transgenic study demonstrate that the high-protein *POWR1* allele can be employed to address the worldwide demand for high-protein food and feed using both breeding and biotech approaches. The study provides insight into the molecular and genetic basis of the vital inter-correlated seed traits and their role in domestication, which should be significant for developing effective strategies to improve the seed quality and yield in soybean and other seed crops.

## Results

**A TE insertion in *POWR1* is associated with seed oil, protein content, and seed weight**. We conducted genome-wide association studies (GWASs) using GLM and MLMM models with 38,066 genome-wide SNPs (Single Nucleotide Polymorphisms) and identified three loci on chromosomes 10, 11, and 20 that were significantly associated with oil content with $\alpha$ values <0.05 in a panel of 278 diverse soybean accessions (Supplementary Figs. 1 and 2; Supplementary Data 1). The most significant SNP (ss715637321 on chr20: 32,835,139) coincided with a genomic region where large-effect protein and oil QTLs have been repeatedly mapped over the last three decades[6,9,11–14]. Our current analysis focused on the QTL on chr20 and delimited it to a 4-Mb interval (chr20: 29,050,000–33,120,000) (Supplementary Fig. 2). To uncover the underlying causative DNA variant, we analyzed whole-genome resequencing data of the 278 accessions. The association studies with the SNPs and InDels (Insertions and Deletions) present in the 4-Mb region identified a prominent cluster of 25 significant associations for oil, which spanned a 154-kb region (chr20: 31,658,904–31,812,853) (Fig. 1a). Out of the 25 highly significant oil-associated DNA variants (23 SNPs and 2 InDels with $p \leq 1 \times 10^{-17}$), a 321-bp InDel showed the most significant association ($p = 6.17 \times 10^{-24}$) (Fig. 1a). The 321-bp InDel was also among the significant associations with protein content and 100-seed weight in the association analyses at a single nucleotide resolution (Fig. 1a and Supplementary Data 2). Moreover, except the 321-bp InDel present in the coding region of *Glyma.20G085100*, none of the other traits-associated DNA variants were in the coding regions of the 12 genes found across the identified 154-kb region (Fig. 1b, c and Supplementary Data 2).

We next examined the effects of the InDel on seed oil and protein content, and seed weight in the panel by dividing it into *G. max*-Del, *G. max*-Ins, and *G. soja*-Del accessions. Interestingly, we did not observe any *G. soja* accession containing the insertion allele in the panel. However, both Del-carrying *G. soja* and *G. max* accessions were significantly lower in oil (by 8.2% and 7.1%, respectively) and 100-seed weight (by 14.0 and 14.59 g, respectively), but higher in protein (by 5.1 and 7.3%, respectively) than *G. max*-Ins accessions. In contrast, no or relatively small differences (on average 1.5% for oil, 2.2% for protein, and 0.2 g for 100-seed weight) were observed between *G. soja*-Del and *G. max*-Del accessions for the three seed traits. This suggests that the observed phenotypic differences may be primarily due to the InDel variation rather than the differences between populations of *G. max* and *G. soja* (Fig. 1d). These results further support that the QTL on chr20 is associated with seed oil, protein, and seed weight and the 321-bp InDel in *Glyma.20G085100* is the causative variant for the chr20 QTL.

BLAST (Basic Local Alignment Search Tool) revealed that the 321-bp InDel sequence in *Glyma.20G085100* was highly homologous to the terminal sequence of a LINE (Long INterspersed Element) transposable element (TE), which belongs to the Gml1 family[15]. We designated this gene, *Glyma.20G085100*, as *POWR1* for seed *Protein, Oil, Weight Regulator 1*. The *POWR1* alleles with and without the 321-bp insertion were named *POWR1*$_{+TE}$ and *POWR1*$_{-TE}$, respectively.

**The TE insertion also underlies the large-effect protein and/or oil QTLs previously mapped to the genomic region on chr20**. We genotyped a bi-parental population of 300 recombinant inbred lines (RILs) generated from Williams 82 [*G. max*, HOLP (High Oil, Low Protein)] and PI479752 [*G. soja*, LOHP (Low Oil, High Protein)] with the SoySNP50K array, and the data obtained were further used for both GWAS (GWAS$_{RIL}$) and linkage mapping. Linkage mapping identified two major QTLs on chr15 and chr20. The QTL on chr20 had a large effect and explained 21.9% of total oil and 23.4% of total protein variation (Supplementary Fig. 3a). GWAS$_{RIL}$ identified three adjacent SNPs at the 154-kb interval on chr20 (ss715637271, ss715637273, and ss715637274) showing the most significant, equal associations ($p = 1.19 \times 10^{-17}$) with oil and protein content (Fig. 1a and Supplementary Fig. 3b, c). The TE insertion was located between two of the three most significant SNPs (ss715637273, ss715637274). Whole-genome sequence analysis and PCR assay results verified the presence of the TE insertion in Williams 82 and absence in PI479752 (Fig. 1c, f) and the complete co-segregation of the TE insertion with high oil and low protein in 30 selected RILs containing either high protein or high oil (Supplementary Table 1). RILs carrying the TE insertion contained 5.2 % higher oil ($p = 4.00 \times 10^{-13}$) and 6.2% lower protein ($p = 3.53 \times 10^{-10}$) than those RILs lacking the insertion (Supplementary Table 1). GWAS$_{RIL}$ and linkage mapping provided additional evidence supporting that the TE insertion is the causative variant underlying the oil and protein QTL on chr20.

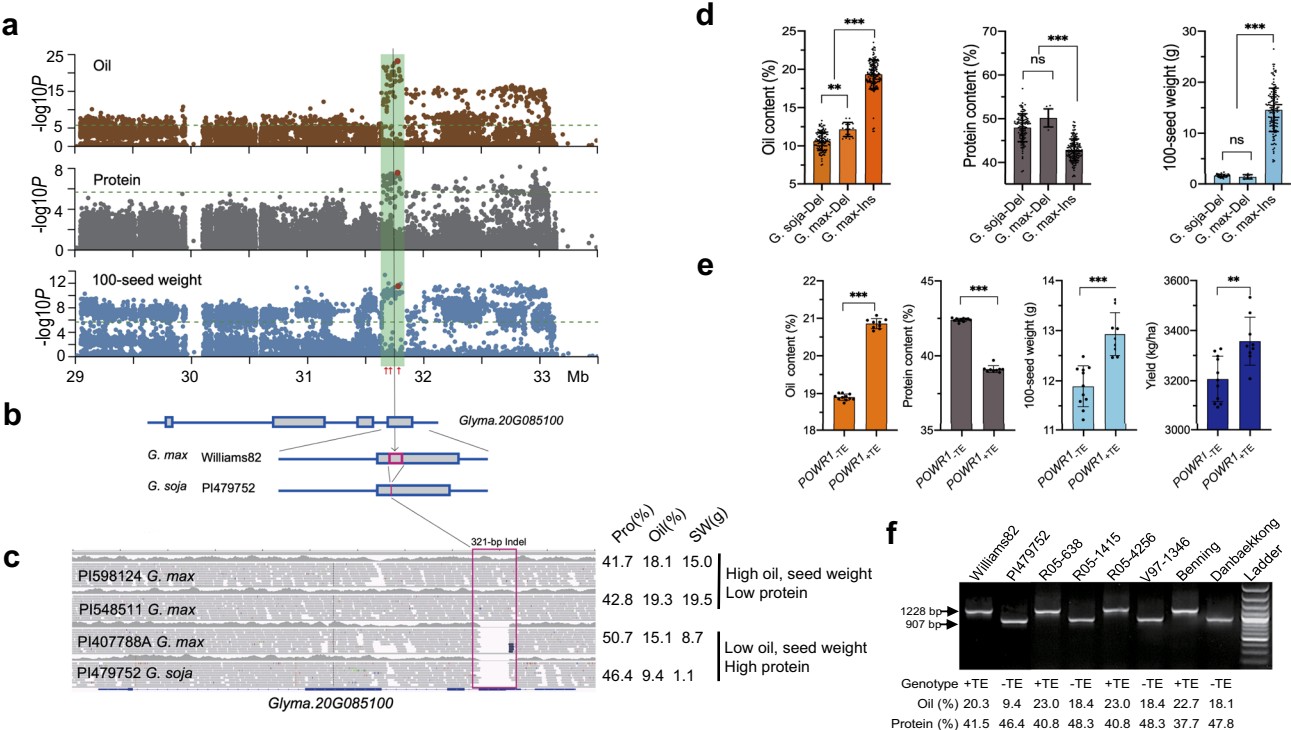

**Fig. 1 Identification of the causative gene and allele underlying a large effect oil, protein, and seed weight QTL on chr20. a** Manhattan plots illustrating associations of oil, protein content, and 100-seed weight with SNPs and InDels across the 4-Mb interval on chr20. The 154-kb region is highlighted in green. The 321-bp InDel (red dots) and the significance threshold after Bonferroni correction (horizontal dotted lines) are shown. The red arrows at the bottom of the panel mark the locations of the three most significantly associated SNPs identified in GWAS$_{RIL}$. **b** Gene structure of *Glyma.20G085100* with and without the 321-bp InDel (red box). **c** Sequencing read alignments of *Glyma.20G085100* from two high oil/low protein accessions and two low oil/high protein accessions showing the 321-bp InDel. Seed oil (Oil), protein (Pro) content, and 100-seed weight (SW) of each accession are shown. **d, e** Allelic effect of the 321-bp InDel on oil, protein, 100-seed weight, and yield among the association panel **d** (*n* = 116, 8, and 154 for *G. soja*-Del, *G. max*-Del, and *G. max*-Ins, respectively) and among NILs (**e**) (*n* = 12). Trait values are shown as the mean ± SD (standard deviation). Statistical significance was determined by one-way ANOVA for the association panel and by two-tailed unpaired Student's *t* test for the NILs. *$p < 0.05$, **$p < 0.01$, and ***$p < 0.001$. Trait values are shown as the mean ± SD (standard Derivation). **f** PCR genotyping assay for the TE insertion in parental lines of four RIL populations. PCR amplicons of 1228 bp and 907 bp represent the presence and absence of the TE insertion, respectively. Genotypes for TE insertion, and oil and protein content for each accession are shown. This experiment was repeated twice. Source data are provided as a Source Data file.

Large-effect protein and/or oil QTLs have been identified in the genomic regions containing *POWR1* in multiple bi-parental RIL mapping populations. We further genotyped the TE insertion in parents of 15 mapping populations previously used for protein and/or oil QTL mapping[6,9,11–13]. The parents of seven bi-parental mapping populations (3 *G. max* × *G. soja*, 4 *G. max* × *G. max*) were polymorphic for the TE insertion while the parents of the other eight populations (*G. max* × *G. max*) were not (Fig. 1f and Supplementary Data 3). Notably, the oil and/or protein QTL in the genomic regions on chr20 was only identified in populations derived from parents polymorphic for the TE insertion but not in populations whose parents were not polymorphic. In the seven pairs of parental lines, the high-oil parent carried the TE insertion while the low-oil parent did not (Fig. 1f and Supplementary Data 3). These results suggests that the TE insertion is the causative DNA variant for those QTLs, which were previously mapped to the genomic region on chr20.

***POWR1* is also associated with field yield.** We further investigated the correlation of the TE insertion allele and seed traits by analyzing a set of *G. max* near-isogenic lines (NILs) for the QTL on chr20. The TE insertion polymorphism among NILs highly correlated with the phenotypic variation for seed protein and oil content as well as the 100-seed weight trait (Fig. 1e and Supplementary Fig. 4). The NILs lacking the 321-bp TE (*POWR1*$_{-TE}$)

exhibited a significant and consistent increase in seed protein (3.29%, $p < 0.001$) and decrease in seed oil content (1.95%, $p < 0.001$) and a marked reduction in 100-seed weight (1.04 g, $p < 0.001$) compared to those with the TE insertion (*POWR1*$_{+TE}$) (Fig. 1e). Importantly, the *POWR1*$_{+TE}$-carrying lines had 150.3 kg/ha higher yield than *POWR1*$_{-TE}$ lines ($p < 0.01$). This suggests that *POWR1*$_{+TE}$ plays an important role in increasing field yield potential in addition to the regulation of the three seed-related traits (oil, protein, and seed weight).

**The TE insertion truncates the CCT domain of POWR1 and alters its nuclear localization.** *POWR1*$_{-TE}$ encodes a protein containing a highly conserved CCT (<u>C</u>ONSTANS, <u>C</u>O-like, and <u>T</u>OC1)-domain at the C-terminus (Fig. 2a)[16]. *POWR1*$_{-TE}$ in wild soybean PI479752 contains an intact CCT domain, whereas *POWR1*$_{+TE}$ in cultivated soybean Williams 82 contains the TE insertion in exon 4 that encodes a part of the CCT motif (Fig. 2a–c). The LINE transposon in *POWR1*$_{+TE}$ is 304 bp in size and generated a 17-bp target site duplication (GTATGCTTGCCGCAAAA) upon insertion (Fig. 2c). Consequently, this 304-bp insertion resulted in a reading frameshift mutation and produces a protein containing a truncated CCT domain of 27 amino acids and a distinct amino acid sequence at the C-terminus compared to POWR1$_{-TE}$ (Fig. 2b). POWR1 homologs were identified in both dicot and monocot species

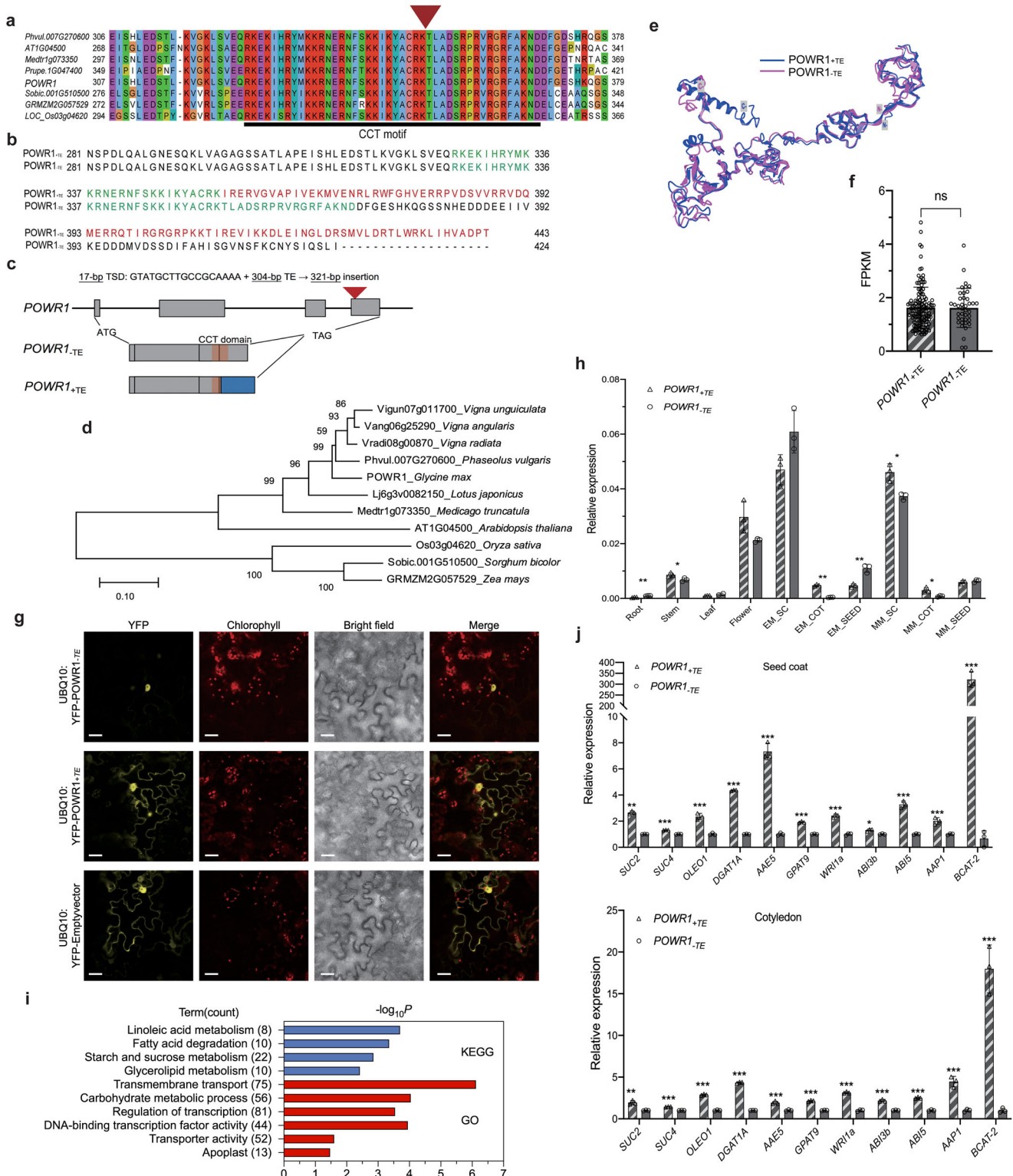

suggesting its ancient origin in plants (Fig. 2a, d). None of the closely related CCT genes among the examined legume species have the TE insertion (Fig. 2a, d). LINE transposons do not require excision to replicate. The mutation generated by the insertion should be stable[15]. This indicates that the TE insertion has occurred and plays a lineage-specific role in soybean.

The TE insertion causes an overall small structural change in the predicted 3D protein structure of POWR1$_{+TE}$ compared to POWR1$_{-TE}$ except at the C-terminal end harboring the CCT domain (Fig. 2e). Interestingly, the second half of the CCT-motif contains a putative nuclear localization signal that is affected by the structural change. We examined the subcellular localization of POWR1$_{-TE}$ and POWR1$_{+TE}$ to determine any alteration in their subcellular localization. The transient expression of the two protein alleles in tobacco (*Nicotiana benthaminana*) leaves revealed that POWR1$_{-TE}$ was exclusively localized in the nucleus (Fig. 2f). This indicates that POWR1 is a transcription-associated factor consistent with the fact that many CCT-domain proteins

**Fig. 2 Gene structure, subcellular localization, and expression of POWR1. a** Sequence alignment of POWR1 orthologs in divergent plant species showing the conserved CCT domains. The TE insertion is marked by a red arrow. **b** C-terminal sequence alignment of POWR1$_{+TE}$ and POWR1$_{-TE}$ showing the conserved CCT domain sequence (in green) and altered C-terminal sequence of POWR1$_{+TE}$ (in red). **c** Gene structure of *POWR1* with and without the 321-bp TE insertion. The position of the TE insertion is shown by a red arrow. The CCT domain and the mutated C-terminal sequence are shown in orange and blue, respectively. The 17-bp target site duplication (TSD) and the TE size are shown. **d** A phylogenetic tree of POWR1$_{-TE}$ orthologs from monocot and dicot species. **e** Superimposed predicted structures of POWR1$_{-TE}$ and POWR1$_{+TE}$ showing almost identical N-termini but distinct C-termini. **f** Subcellular localization of POWR1$_{-TE}$ and POWR1$_{+TE}$ in tobacco cells. Scale bar = 20 μm. This experiment was repeated twice. **g** Similar expression patterns of *POWR1$_{-TE}$* and *POWR1$_{+TE}$*. Y-axis indicates the expression levels relative to *GmCYP2*. EM: Early Maturation Stage; MM: Mid-Maturation Stage; SC: Seed Coat; COT: Cotyledon. Statistical significance is determined by two-tailed unpaired Student's *t* test. *p* values = 0.0003, 0.0312, 0.0697, 0.0659, 0.0633, <0.0001, 0.0023, 0.0103, 0.0190, 0.41048. *$p < 0.05$, **$p < 0.01$, and ***$p < 0.001$. Expression levels are shown as the mean ± SD. **h** Comparable expression levels of *POWR1$_{-TE}$* and *POWR1$_{+TE}$* in seeds of 40 *POWR1$_{-TE}$* accessions and 132 *POWR1$_{+TE}$* accessions at mid-maturation stages. **i** GO and KEGG terms enriched for the differentially expressed genes between *G. max* accessions containing *POWR1$_{-TE}$* and *POWR1$_{+TE}$*. P value (X-axis) and the number of observed genes (in parentheses) for each term are shown. **j** Relative expression levels of selected genes in seed coat and cotyledon of NILs containing *POWR1$_{-TE}$* or *POWR1$_{+TE}$* ($n = 3$). Statistical significance was determined by two-tailed Student's *t* test. Top: *p* values = 2.22e-05, 1.18e-05, 0.0008, 2.62e-08, 6.88e-05, 1.59e-05, 0.0008, 0.0046, 0.0001, 0.0026, 0.0013. Bottom: 0.0023, 2.78e-05, 3.12e-06, 2.45e-07, 0.0013, 1.34e-05, 1.07e-05, 8.41e-05, 5.51e-5, 1.92e-05. Source data are provided as a Source Data file.

are transcription co-factors[17]. However, POWR1$_{+TE}$ was localized in both nucleus and cytoplasm as the empty vector. This implies that the CCT domain is a functional element for its subcellular localization. The TE insertion may either alter or inactivate the function of POWR1 leading to the disruption of its exclusive nuclear localization.

***POWR1 is specifically expressed in the seed coat and flowers.*** Gene expression analysis revealed that both *POWR1* alleles are preferentially expressed in flowers and developing seed coat at the early and middle maturation stages. Both alleles showed little to no significant expression differences in flowers and seed coats (Fig. 2g). In addition, *POWR1$_{+TE}$* and *POWR1$_{-TE}$* did not exhibit any significant difference in their expression in mid-maturation seeds based on transcriptome analyses of 132 soybean accessions containing *POWR1$_{+TE}$* and 40 containing *POWR1$_{-TE}$* (Fig. 2h). We also did not observe sequence variation in the 2-kb promoter sequences between *POWR1$_{+TE}$* in Williams 82 and *POWR1$_{-TE}$* in PI479752, the parental lines of the RIL population analyzed above (Supplementary Fig. 5). Thus, both gene expression and sequence comparisons suggest that the TE insertion caused little to no expression change in flowers and seed coats. The preferential expression of *POWR1* in the seed coat may imply its possible involvement in nutrient transport, the integral physiological role of the seed coat. Real-Time PCR showed small expression differences with statistical significance in some low-expressing tissues such as the embryo at the early maturation stage. It remains to be determined if these small expression differences significantly affect the seed traits.

***POWR1 preferentially regulates lipid metabolism and nutrient transport genes.*** To gain insight into the molecular mechanism underlying regulatory function of *POWR1* on the seed traits, we compared the transcriptomes of mid-maturation seeds between four and six *G. max* accessions carrying *POWR1$_{-TE}$* and *POWR1$_{+TE}$*, respectively. As expected, the two genotypic groups have no significant difference in *POWR1* expression (Supplementary Table 3). However, the transcriptome comparison identified a total of 1163 differentially expressed genes (DEGs) associated with the TE insertion (Supplementary Data 4). KEGG and GO terms related to fatty acid, lipid, starch, and sucrose metabolism, transmembrane transport, carbohydrate metabolism, regulation of transcription (biological process) and apoplast (cellular component) were significantly enriched for the DEGs (Fig. 2i). This result corresponds with the preferential expression of *POWR1* in seed coat tissues, which are mainly responsible for transporting nutrients to support metabolic activities in the

cotyledons during seed development[18] (Fig. 2g). It is also consistent with the pleiotropic effect of *POWR1* on oil and protein content and seed weight.

A set of regulatory and metabolic genes involved in protein and oil production were differentially regulated by the TE insertion in the seed coat and/or cotyledon at mid-maturation stage (Fig. 2j). For example, lipid biosynthesis genes (*DGAT1*, *AAE*, *GAPT9*) and sugar transporter genes (*SUC2*, *SUS4*) showed significantly increased expression in *POWR1$_{+TE}$* relative to *POWR1$_{-TE}$* background in both seed coat and cotyledon tissues. The most striking increase in expression level was observed for *BCAT2*, which is involved in branched-chain amino acid metabolism, suggesting its contributing role to relatively lower protein content in *POWR1$_{+TE}$* than in *POWR1$_{-TE}$*. It is known that the regulators (*WRI1*, *ABI3b*, and *ABI5*) are involved in seed development and size as well as oil accumulation[19–21]. The regulatory genes were also upregulated in *POWR1$_{+TE}$* relative to that in *POWR1$_{-TE}$*. This is suggestive of the regulators acting downstream of *POWR1*. Their differential expression in NILs suggest that they are likely part of the transcriptional regulatory cascade underlying *POWR1* regulation for seed traits.

***Expression of POWR1$_{-TE}$ increases protein content and reduces oil content and seed weight in transgenic plants.*** The intact *POWR1$_{-TE}$* cDNA driven by a strong and constitutively expressing Ubiquitin promoter or the 1.9-kb *POWR1* native promoter were introduced into *POWR1$_{+TE}$ G. max* background (cultivars Maverick and Williams 82, respectively) to validate its function. Two overexpressing (OE) Ubiquitin promoter-driven *POWR1* transgenic events (UbiOE1 and 2) showed a high-level expression of *POWR1$_{-TE}$* based on qRT-PCR analyses (Supplementary Fig. 7e). The UbiOE1 and UbiOE2 seeds contained significantly higher protein content (2.50%, $p < 0.01$) but showed reduced oil content (2.36%, $p < 0.05$) and 100-seed weight (3.57 g, $p < 0.05$) compared to the non-transgenic control seeds (Fig. 3a). Analysis of 18 independent T1 transgenic plants showed that the lines with native promoter-driven *POWR1$_{-TE}$* (Nat-OE) contained significantly higher seed protein content (4.39%) and lower seed oil (1.31%) but no significant change was observed for seed weight (Fig. 3b). These results support that *POWR1* controls soybean seed oil and protein content and seed weight and can be manipulated to alter these traits for seed quality improvement in soybean.

***POWR1 is a domestication gene.*** The distribution of *POWR1* alleles was studied next in an expanded soybean population consisting of 548 diverse accessions. Principal component

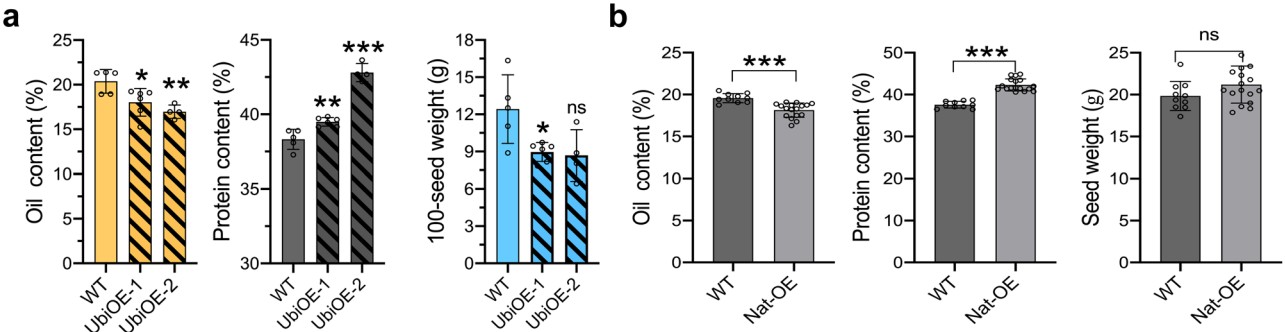

**Fig. 3 Seed oil and protein content and weight in transgenic soybean overexpressing (OE) POWR1_-TE. a** Seed protein, oil, and weight of T2 plants from each of the two transgenic events containing Ubi-promoter driven POWR1_-TE cDNA (Ubi-OE) ($n = 4–6$). Unpaired $t$-test $p$ values = 0.025, 0.0026 for oil content; 0.0036, <0.0001 for protein content; 0.0158, 0.0604 for seed weight. **b** seed protein, oil content, and 100 seed weight of T1 plants containing native promoter driven POWR1_-TE cDNA (Nat-OE) from 18 independent transgenic events ($n = 18$). Unpaired $t$-test $p$ values = <0.0001 for oil content, <0.0001 for protein content, 0.1095 for seed weight. Statistical significance was determined by two-tailed unpaired Student's $t$ test. Source data are provided as a Source Data file.

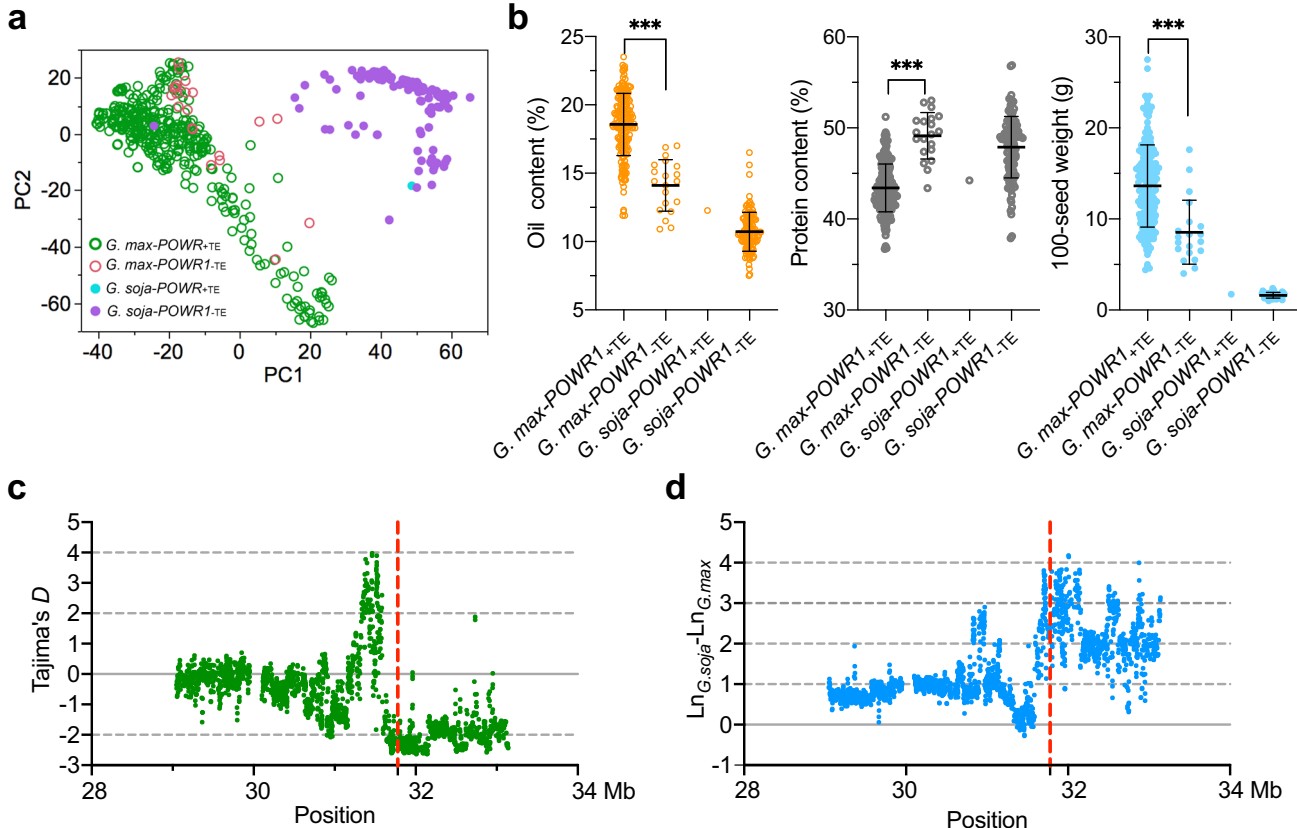

**Fig. 4 Distribution and diversity analysis of POWR1 alleles in soybean. a** PCA of 548 cultivated and wild soybean accessions with different allele types is shown. **b** Comparison of seed oil and protein content and 100-seed weight of G. max and G. soja accessions carrying POWR1_+TE or POWR1_-TE ($n = 377$; 21, 1,149). Statistical significance was determined by two-tailed unpaired Student's $t$ test. *$p < 0.05$, **$p < 0.01$, and ***$p < 0.001$. Unpaired $t$-test $p$ values = <0.0001, 0.2775 for oil content; <0.0001, 0.2823 for protein content; <0.0001, 0.7034 for seed weight. Trait values are shown as the mean ± SD. **c, d** Tajima's $D$ and Ln($\pi$-G. soja)-Ln($\pi$-G. max) between G. max and G. soja accessions across the 4.1 Mb region containing POWR1. The vertical solid red line indicates the physical position of POWR1. Source data are provided as a Source Data file.

analysis (PCA) revealed that the majority of the 150 G. soja accessions were clustered together as one group separately from the group containing 398 G. max accessions (Fig. 4a). The examination of the POWR1 allelic distribution across the cultivated and wild natural population revealed a nearly complete association of POWR1_-TE and POWR1_+TE alleles with G. soja and G. max populations, respectively, with a few exceptions. Specifically, 94.7% (377 of 398) of G. max possessed the

POWR1_+TE allele, while all G. soja except one (149 of 150) carried the POWR1_-TE allele (Fig. 4a). As expected, the POWR1_-TE allele was associated with 4.47% lower oil and 5.73% higher protein contents, and 5.08 g lower seed weight than the POWR1_+TE allele in G. max accessions ($p < 0.001$). This pattern of allelic effects on the seed traits was observed to be consistent across G. soja groups (1.56% for oil, 3.65% for protein, and 0.12 g for seed weight) (Fig. 4b). A genomic scan revealed that POWR1 is located within

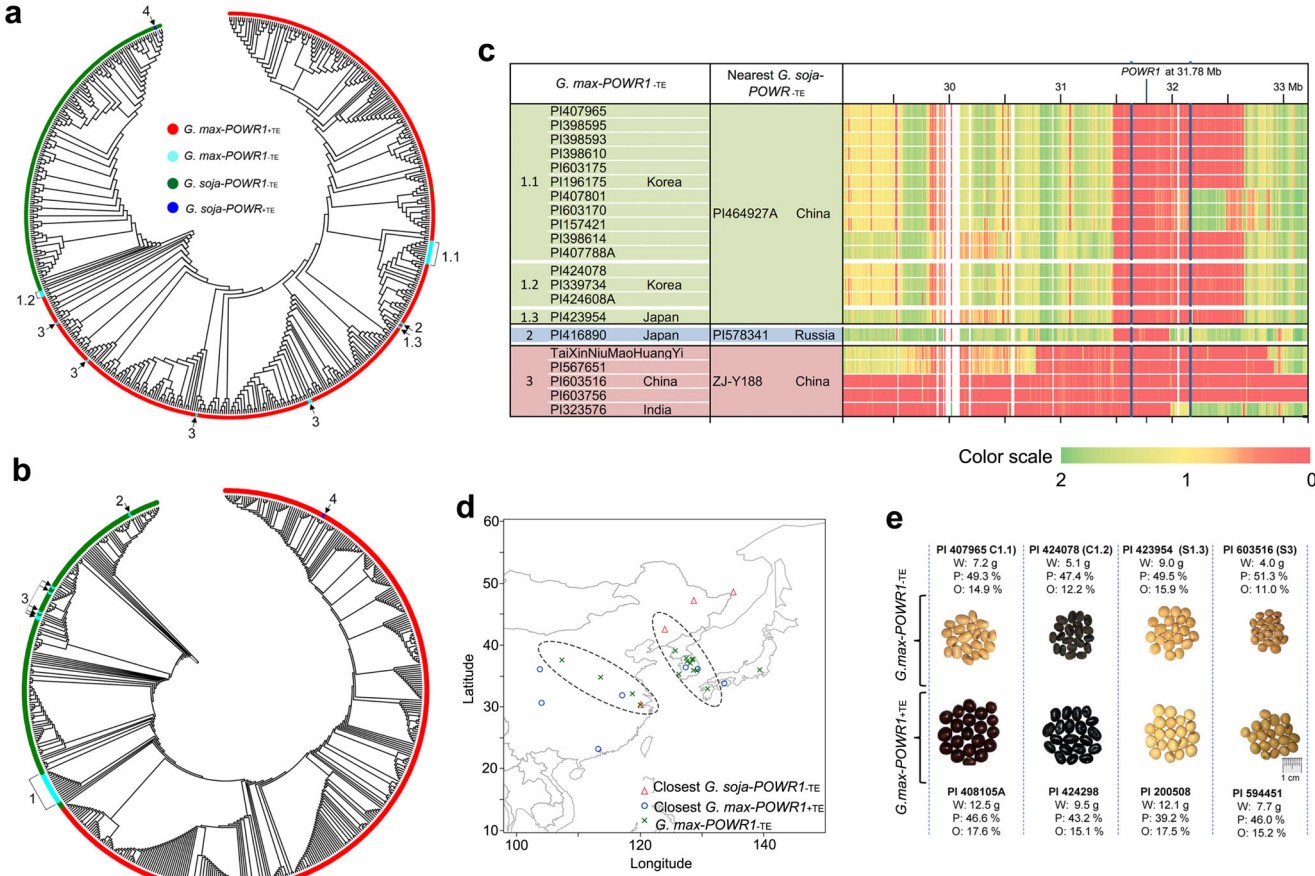

**Fig. 5 Dynamic interspecific transfer of *POWR1*. a, b** A global (**a**) and a local (**b**) phylogenetic tree constructed from 548 *G. soja* and *G. max* accessions using genome-wide SoySNP50K SNPs and 1000 SNPs spanning the 154-kb region containing *POWR1*, respectively. Labels (1, 2, 3, 4) in the local tree indicate four clusters of accessions containing unusual genotypes [*G. max-POWR1*$_{-TE}$ (1,2,3), *G. soja-POWR1*$_{+TE}$ (4)]. The labels in the global tree correspond to the labels in the local tree. **c** Pairwise nucleotide distance analyses of each *G. max-POWR1*$_{-TE}$ accession with its closest *G. soja-POWR1*$_{-TE}$ accession across the 4-Mb region. The clusters of these accessions and the countries of their origins are shown. The pairwise distance is indicated by a color scale from red (close) to green (distant). The domestication region identified based on Tajima's *D* (see Fig. 4c) is delineated by blue lines. The physical location of *POWR1* is marked on top. **d** Interspecific allele transfer. Geographic origins of *G. max-POWR1*$_{-TE}$ accessions and their closest *G. soja-POWR1*$_{-TE}$ and *G. max-POWR1*$_{+TE}$ accessions are shown. Dotted circles depict the geographic regions in China, North and South Korea, and Japan where the interspecific allele transfer might occur. Latitude and longitude are shown. **e** *G. max-POWR1*$_{-TE}$ accessions have smaller seeds than their closest related *G. max-POWR1*$_{+TE}$ accessions. The top panel shows representative *G. max* accessions from clusters 1.1 (C1.1) and 1.2 (C1.2), 1.3 (S1.3), and 3 (S3) that carry *POWR1*$_{-TE}$. The bottom row shows their closest related *G. max-POWR1*$_{+TE}$ accessions. PI number, 100-seed weight (W), seed protein content (P), and seed oil content (O) are provided for each accession. Source data are provided as a Source Data file.

an approximately 520-kb selective sweep region (chr20: 31,641,057–32,160,913) as inferred by Tajima's *D* of $< -2$ (Fig. 4c) and high *G. soja/G. max* π ln-ratios (larger than 2.4) (chr20: 31,654,290–32,157,761) (Fig. 4d). These results indicate that *POWR1* is a domestication gene and *POWR1*$_{+TE}$ was subjected to artificial selection during soybean domestication which contributed to higher seed weight and oil content in *G. max* compared to *G. soja*.

**Dynamic interspecific transfer of *POWR1* alleles after soybean domestication.** A total of twenty-one *G. max-POWR1*$_{-TE}$ accessions and one *G. soja-POWR1*$_{+TE}$ accession had *POWR1* alleles contrasting to the majority of *G. max-POWR1*$_{+TE}$ and *G. soja-POWR1*$_{-TE}$ accessions (Fig. 4a). To learn about the origin of the unusual presence of *POWR1* alleles in these exceptional accessions, we constructed a global and a local phylogenetic tree using the respective genome-wide SoySNP50K SNPs and the whole genome resequencing-based SNPs present in the 154-kb region among the 548 accessions (Fig. 1a). The global tree resembled the PCA result (Figs. 4a and 5a). All *G. max* accessions

[*G. max-POWR1*$_{+TE}$ and *G. max-POWR1*$_{-TE}$ (clusters 1.1, 1.2, 1.3, 2, 3)] clustered together and were separated from all *G. soja* accessions [*G. soja-POWR1*$_{-TE}$ and *G. soja-POWR1*$_{+TE}$ (singleton 4)] regardless of the TE insertional variation (Fig. 5a). However, in the local phylogenetic tree, all *G.max-POWR1*$_{-TE}$ accessions changed from the *G. max* cluster as seen in the global tree to the more diverse *G. soja* clusters (clusters 1, 2, 3) while the *G. soja-POWR1*$_{+TE}$ accession (singleton 4) switched to the *G. max* cluster (Fig. 5a, b). It is indicative of the transfer of *POWR1* alleles between *G. soja* and *G. max* after domestication resulting in the *G. soja-POWR1*$_{+TE}$ and the *G. max-POWR1*$_{-TE}$ accessions. Excluding these accessions with post-domestication allele transfer, all remaining *G. soja* accessions carried *POWR1*$_{-TE}$ and all *G. max* accessions contained the *POWR1*$_{+TE}$ allele. We also observed the same distribution pattern of the two *POWR1* alleles among the 3,956 soybean accessions (3451 *G. max* accessions and 363 *G. soja* accessions) that have been sequenced recently (Supplementary Fig. 9)[22–26]. The complete association of *POWR1*$_{+TE}$ and *POWR1*$_{-TE}$ with *G. max* and *G. soja* accessions, respectively, and its function in controlling seed weight and yield, an

important domestication syndrome[27], support the hypotheses that $POWR1_{+TE}$ was subjected to a strong and exclusive selection during domestication and is a player in soybean domestication. The local tree analysis clearly clustered all the *G. max-POWR1_{-TE}* accessions into three clusters (clusters 1, 2, 3) of *G. soja* clade. However, these accessions were split into different, distant clusters (cluster 1 to 1.1, 1.2, 1.3 and scatted distribution of cluster 3 accessions) of *G. max* clade as per the global tree. This suggests that the fragments harboring $POWR1_{-TE}$ were transferred into diverse *G. max* accessions likely from *G. soja* accessions, hence producing those *G. max-POWR1_{-TE}* accessions with diverse genetic backgrounds, as shown by their scatted distribution in the global tree (Fig. 5a).

Estimation and plotting of pairwise genetic distance between each of the 21 *G. max-POWR1_{-TE}* accessions and their phylogenetically closest *G. soja* accessions (PI464927A, PI578341, and Zj-Y188) in the local tree was carried out across the 4-Mb region to detect the possible transfer of genomic regions harboring $POWR1_{-TE}$ (Fig. 5c). The pairwise distance analysis showed diverse patterns of highly identical sequences of variable lengths within the region among the three clusters. Briefly, a region (roughly 1.2 Mb long) with high sequence identity either sharing one end or both ends was identified in cluster 1. However, cluster 3 contained the transferred segment carrying the $POWR1_{-TE}$ of variable lengths, whereas cluster 2 showed the shortest transferred fragment containing the $POWR1_{-TE}$ (~500 kb long). These results support that the $POWR1_{-TE}$ in these *G. max* accessions most likely originated from post-domestication allele transfer events and went through multiple chromosomal crossovers. The mapping of these accessions to their geographic origins further revealed close geographical proximity of *G. max-POWR1_{-TE}* with their phylogenetically closest *G. soja-POWR1_{-TE}* (in the local tree) with multiple geographic locations (South Korea, Japan, China) of East Asia (Fig. 5d). These imply that the allele transfers occurred most likely within these regions. Indeed, those *G. max-POWR1_{-TE}* from East Asia contained 6.5% higher protein content than their closed-related *G. max-POWR1_{+TE}* accessions despite an average decrease of 2.7% oil content and 3.2 g 100-seed weight in soybean (Fig. 5f and Supplementary Fig. 6). This is beneficial as it aligns with the needs for high-protein soy food in East Asia.

## Discussion

Large-effect protein and/or oil QTL(s) have been repeatedly mapped to a genomic region on chromosome 20 in multiple studies. The QTLs for multiple agronomically important traits including seed weight and yield were also observed in the region[6,9,11–14,28]. Significant efforts have been dedicated to identifying causative gene(s) and variant(s) underlying the QTLs on chr20 during the past three decades. By leveraging the whole-genome re-sequencing data of 278 highly diverse accessions, a single nucleotide level high-resolution trait association analysis was coupled with high-confidence bi-parental genetic mapping in this present study. This enabled uncovering of not only a single gene, POWR1 that underlies the previously reported QTLs regulating multiple seed traits but also detection of a TE insertion in POWR1 as the causative allele (Fig. 1). The causality is further supported by its functional characterization of transgenic soybean lines, subcellular localization studies of the two POWR1 alleles and gene expression analyses. Association of $POWR1_{+TE}$ with higher yield at field level was observed in near isogenic lines which might be due to the regulation of the positively correlated seed weight, which is a yield component[29]. Although the observed pleiotropic effect of POWR1 on the important seed traits may pose a challenge to improve all traits simultaneously, it offers an opportunity to use appropriate alleles for enhancing a specific trait or a combination of the traits. For example, here we demonstrate that both transgenic expression of the high-protein $POWR1_{-TE}$ allele in elite lines carrying a $POWR1_{+TE}$ allele and transfer of $POWR1_{-TE}$ allele from *G. soja* into Asian *G. max* accessions lead to a higher level of seed protein content. Thus, the high-protein allele could be an excellent gene target for developing high-protein cultivars using molecular breeding and biotechnology[28]. Further understanding of its underlying molecular mechanisms potentially provides strategies to uncouple the antagonistic relationship of protein with seed oil content, seed weight, and yield.

It has been shown that CCT domain-containing genes mainly function in photoperiod-related adaptation in *Arabidopsis* and cereals[17]. However, this study demonstrates that POWR1, a CCT domain-containing gene, plays a role in the regulation of oil and protein content, and seed weight/yield in soybean. In consistency with its function, POWR1 is shown to be preferentially expressed in the seed coat, a tissue that is a key player in nutrient transport to cotyledon for storage reserve production and seed filling[30]. The TE insertion in the CCT domain disrupted the exclusive localization of POWR1 in the nucleus but caused little or no change in its expression in seeds and other seed compartments and tissues. The TE insertion that increased oil and seed weight therefore is most likely to function through altering its protein structure and not its expression. Given the role of the CCT domain in DNA binding and protein interaction[31], our transcriptome and real-time RT-PCR analyses showed that POWR1 is likely to regulate the expression of genes involved in oil and protein metabolism, nutrient transport, and regulation of seed development. For example, ABI5 which is well-known for determining seed size[19] and BCAT2 with a function in protein degradation[32] had significantly higher expression in a $POWR1_{+TE}$ background compared to a $POWR1_{-TE}$ background. This is in accordance with the result that seeds carrying $POWR1_{+TE}$ contain lower protein and higher oil content and higher seed weight. Based on these observations, it can be hypothesized that $POWR1_{-TE}$ may act upstream of the metabolic genes, transporter genes and regulators (including WRI1a, ABI5), which collectively affect the three vital seed traits.

A nearly complete fixation of $POWR1_{+TE}$ in *G. max* and absence of $POWR1_{+TE}$ in *G. soja* were observed among the population of 548 accessions used in the study and among the 3,956 soybean accessions that have been sequenced recently (Supplementary Fig. 9)[22–26,33]. The TE insertion in POWR1 therefore may be among the key events during the transition from *G. soja* to *G. max*. Selection for larger seed soybean exhibiting higher seed yield and seed oil could be the cause of fixing the $POWR_{+TE}$ in modern *G. max*. It is unlikely that seed oil as a non-visible seed quality trait was strongly selected during early soybean domestication. Thus, oil increase could simply be the by-product as it is pleiotropically controlled by $POWR1_{+TE}$ (Fig. 6). The larger seed size trait has been suggested as an early domestication trait in several cereal crops[34] and is most likely true for soybean as well. Interestingly, recent archeological studies suggest that the increase in seed oil content during domestication might have occurred no later than seed enlargement in soybean[35,36]. In other words, seed oil increase might have occurred simultaneously with seed size enlargement, which is consistent with POWR1 pleiotropically controlling both traits. The resultant simultaneous decrease in seed protein content due to the preferential selection of $POWR1_{+TE}$ might not have had significant impact on ancient agriculture. However, at present-day it is a major concern of the soybean industry as the compromised seed protein content is not desired for the animal feed industry and the demand for plant protein for human consumption is also on the

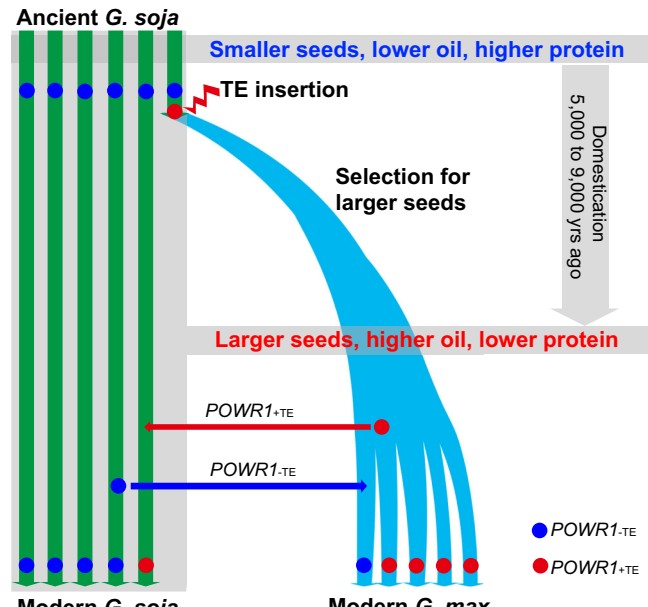

**Fig. 6 A proposed model of POWR1 as a player in soybean domestication.**
The insertion of a LINE transposon represents an important event in the transition from *G. soja* to *G. max* during soybean domestication. Following the TE insertion, the progeny from plants containing $POWR1_{+TE}$ expands due to selection for bigger seeds by ancient farmers. Selection for larger seeds results in complete fixation of $POWR1_{+TE}$ in all *G. max* accessions, which contributes to shaping cultivated soybean with increased oil but reduced protein content in seeds in comparison with *G. soja* accessions due to the pleiotropy of *POWR1* on the seed traits. Transfer of *POWR1* alleles between *G. max* and *G. soja* also occurs after domestication, which result in the *G. max* accessions containing $POWR1_{-TE}$ and one *G. soja* accessions containing $POWR1_{+TE}$. The transfer of $POWR1_{-TE}$ of *G. soja* to *G. max* is likely driven by local needs for high-protein soybeans in East Asia. The one *G. soja* accession with TE insertion likely originated from a hybridization event between *G. max* and *G. soja* as previously reported for semi-wild soybean genotypes[41].

rise[37]. As the low-protein allele is fixed in *G. max*, transfer of the high-protein allele ($POWR1_{-TE}$) from *G. soja* into *G. max* may increase the seed protein content as needed. This represents a reversal of the domestication process, and introgression and selection for $POWR1_{-TE}$ can be seen in Asian breeding programs which are likely to be driven by the stronger need for high-protein soybean in Asia[4,38]. We also observed a soybean accession with TE insertion that was annotated as *G. soja*. However, this accession may be the result of a hybridization event between *G. max* and *G. soja*. Given the outcrossing rate of up to 19% for *G. soja*[39] and up to 6 % for *G. max*[40], natural gene flow and introgression from cultivated soybean to their wild relatives may be a common phenomenon leading to semi-wild soybean (Fig. 6)[41].

This study provides strong evidence that *POWR1* is a player of soybean domestication and pleiotropically regulates seed protein, oil, weight as well as seed yield. The phenotypic value of each seed trait analyzed in the study is the accumulative effect of their QTLs across the soybean genome. Multiple causal genes for each seed trait including *SWEET10a* (also named as *SWEET39*) underlying a major protein and oil QTL have been identified[42–44]. The interaction of *POWR1* with other QTLs that determine the phenotypic value of the seed traits, however, are still largely unknown. In addition, the selection process of domestication genes relative to *POWR1*, which together shaped the cultivated soybean, are still obscure. The comprehensive investigation of

these loci and their relationship with *POWR1* therefore would enable better understanding of the domestication process and the molecular mechanism controlling the correlated seed traits in soybean, which is important in designing effective strategies for soybean seed quality and yield improvement.

## Methods

**Plant materials.** A panel of 548 soybean accessions (398 cultivated soybean *G. max* and 150 wild soybean *G. soja* (Siebold & Zuccarini)) from the genetic resources information network (GRIN) of U.S. National Plant Germplasm System (https://npgsweb.ars-grin.gov/) was used in this study. Out of 548 accessions, 278 accessions (116 *G. soja* and 162 *G. max*) exhibiting variations for seed oil (7.5–23.5%), seed protein (36.7–56.9%), and 100-seed weight (1.0–26.5 g) were used for association analysis (Supplementary Data 1 and Supplementary Fig. 1). An interspecific bi-parental mapping population of 300 $F_{6:7}$ recombinant inbred lines (RILs) derived from a cross between *G. max* cv. Williams 82 and *G. soja* PI479752 was used for genetic linkage mapping. While seed oil content varied from 9.82–20.47%, protein content ranged from 37.64–47.99% among the RILs. Seeds of the parents and RILs, each with two replications, were planted in a randomized block design at the USDA-ARS farms in Beltsville, Maryland during the years 2012 and 2015. The highly homozygous (>99%) near-isogenic lines (NILs) were generated from a $F_7$ plant heterozygous for *POWR1* from a cross of G03-3101 × LD00-2817P[45]. The NILs homozygous at the *POWR1* locus were planted in replications based on randomized complete block design across nine field environments (one in Arkansas, Missouri, North Carolina, and six in Tennessee) during the years 2016 and 2017. The TE variations in the NILs were validated by a PCR assay using a pair of PCR primers flanking the InDel (Supplementary Table 2). All soybean plants including the transgenic lines used for genotyping and quantification of seed traits were grown in the Donald Danforth Plant Science Center greenhouses (St. Louis, Missouri).

**Measurement of seed traits.** Phenotypic data including seed protein and oil content (%), and 100-seed weight (g) for the panel of 548 accessions were acquired from the Germplasm Resources Information Network (GRIN, https://www.ars-grin.gov/). Oil and protein content of the RIL population, the transgenic plants, and all other soybean plants were measured using the near-infrared reflectance (NIR) spectroscopy using a DA 7250 NIR analyzer (Perten Instruments, Sweden) unless specified otherwise. Approximately 50 seeds per line were analyzed and measured twice. For NILs, ~20 g seeds were grounded to powder and also measured with Perten DA 7250 analyzer[45]. Seed trait measurements were averaged over all replications and locations for both NIL groups and compared.

**Sample sequencing, read alignment, and variant calling.** A total of 91 diverse *G. soja* accessions representing over 90% diversity of wild soybeans from the National Plant Germplasm System collection were re-sequenced using the Illumina NextSeq500 sequencer[46]. For the remaining 457 accessions in the association panel and soybean accessions with recently published re-sequencing data, raw sequencing reads were retrieved from the NCBI SRA database (https://www.ncbi.nlm.nih.gov/sra)[22,24–26,33]. All quality-controlled reads were aligned to the *G. max* reference genome (Williams 82.a2 v1)[47] with BWA (0.7.15). DNA variants including SNPs and InDels were called using the GATKs pipeline[48]. The resulting variants were filtered using GATKs VariantFiltration with following parameters: read depth ≥ 5 reads, SNP quality ≥50, and at least 2 SNPs in a 10-bp window were allowed. Read alignments were visualized using the Integrative Genomics Viewer[49]. The resulting 28,708 SNP and 131 InDel markers in a 4-Mb region (29–33.15 Mb) were used to carry out regional association analyses. Whole developing seeds at the mid-maturation stage were collected from plants grown in environmentally controlled greenhouses. Multiple seeds per accession were pooled for transcriptome sequencing[50]. Transcriptome analysis was performed using TopHat (2.0) and Cufflinks (2.2.1)[51], and the FPKMs derived for each sample were normalized based on the quantile method in Cuffdiff[51].

**Association and linkage mapping.** DNA variants were quality controlled before being used for genome-wide or regional association analysis using TASSEL5[52,53] with the following criteria: a maximum minor SNP allele frequency of 0.05, a maximum proportion of heterozygous sites of 0.2, and a minimum number of accessions per site of 85%. Five principal components as determined with TASSEL5 were used as population structure (Q) matrices. Kinship (K) matrix was calculated using centered IBS method in TASSEL5. GLM (general linear model) and MLMM (mixed linear model) were used for genome-wide association mapping and regional association analysis as implemented in TASSEL. For the RIL population, GLM without population structure Q, or GLM with Q, or MLM with Q and kinship K returned almost identical mapping associations for oil and protein using 19,848 SNPs from the SoySNP50K-set[54]. The *Bonferroni*-corrected genome-wide significance threshold was calculated as 0.05/SNP count. Linkage mapping was carried out using Windows QTL Cartographer v2.5[55] and QTLs were detected based on the composite interval mapping with 1000 permutations for each test[56].

**Genetic diversity analyses**. Principal Component Analysis (PCA) of the association panel was conducted in TASSEL5[52] using the SoySNP50K SNPs[54]. Tajima's $D$[57] and the pairwise nucleotide diversity $\pi$[58] were calculated in TASSEL5[52] for the wild and cultivated soybean accessions from the panel. Chromosomal regions accounting for the top 15% ln-ratios (which corresponds to an ln-ratio threshold of about 2.4) or Tajima's $D$ of $< -2$ were considered as domestication regions.

**Phylogenetic tree and sequence alignment analyses**. The unrooted Neighbor-Joining phylogenetic tree of the 548 accessions was constructed using MEGA7[59] with the Maximum Likelihood method based on the Tamura-Nei model[60]. A total of 19,284 genome-wide SNPs were used for the global tree and 1023 SNPs within the 154-kb domestication region were used for the local tree construction. Multiple DNA and protein alignments were performed in Clustal Omega (https://www.ebi.ac.uk/Tools/msa/clustalo/). Protein structures were predicted by I-TASSER[61], compared using RaptorX (TMscore 0.797)[62] and visualized with iCn3D[63].

**RNA extraction and expression analyses**. Soybean NILs for the *POWR1* locus were used for expression analyses. Leaves, roots, and stem tissues were collected at 4 weeks after planting. Fully open flowers were collected after their emergence. Early maturation seeds (25–50 mg weight) and middle maturation seeds (100–125 mg weight) were dissected to obtain seed coat and cotyledon tissues separately. Expression levels of genes of interest were determined based on the BioRad CFX384 Real Time PCR System using SsoAdvanced Universal SYBR® Green Supermix and normalized using *GmCYP2* (*Glyma.12G024700*) expression[43]. Primers for each gene are listed in Supplementary Table 2. Experiments were performed with three biological and technical triplicates. For determining *POWR1* expression levels in soybean transgenic lines, seeds at early maturation (25~50 mg weight) stage were collected. Total RNA was extracted and used for Real-Time PCR and RNA sequencing[50].

**DNA vector construction and soybean transformation**. A vector (backbone pMU106) containing synthetic cDNA of *POWR1*_−TE allele from PI479752 with the Ubi917 promoter, *pUbi:POWR1*_−TE was constructed (Supplementary Fig. 7) and transformed into *G. max* cv. Maverick carrying *POWR1*_+TE using an improved *Agrobacterium* mediated transformation protocol[64]. The presence of the construct in transgenic plants was confirmed by Basta leaf-painting (Supplementary Fig. 7b) and PCR assay (Supplementary Fig. 7c, d). The expression level of *POWR1* in developing seeds at the early maturation stage of transgenic plants was analyzed using the qRT-PCR assay (Supplementary Fig. 7e). The cDNA of *POWR1*_−TE allele along with its 1.9-kb native promoter sequence was cloned into a customized expression vector (backbone pAGM4673) and transformed into soybean cv. Williams 82 (*POWR1*_+TE background) using the *Agrobacterium* mediated transformation at Wisconsin Crop Innovation Center (Madison, WI) (Supplementary Fig. 8). The plants were selected based on spectinomycin resistance followed by PCR confirmation using the primers specific to the vector sequences to determine the positive T0 plants and the primers (F: TATCCATATGACGTTCCAGATTAC GCC; R: ACCTCAGAATTTTGCAGTGTGTGTG) spanning the vector and CDS to identify T1 positive transformants. T1 seeds were measured for total protein, oil and weight. For transient expression, synthesized cDNAs of *POWR1*_−TE and *POWR1*_+TE were cloned into the Gateway entry vector pcr8/Topo. These constructs were moved into plant gateway expression vectors UBQ10:YFP-GW using LR clonase[65,66].

**Transient expression and microscopy analyses**. POWR1_−TE and POWR1_+TE localizations were observed based on transient expression in *N. benthamiana* using the method of Li[67]. Briefly, UBQ10:YFP-POWR1_−TE, UBQ10:YFP-POWR1_+TE, and UBQ10:YFP in *A. tumefaciens* were infiltrated into young leaves of *N. benthamiana* plants (4–6 weeks) using a 3-mL syringe without needle. Leaves were imaged 48 h after infiltration.

Confocal images were obtained with a Leica TCS SP8 confocal microscope using the 63X water immersion lens. Samples were excited with a 514-nm laser line and 649-nm laser line to detect YFP and chlorophyll signals, respectively. Fluorescence emission was collected for best signals of indicated fluorescent probes.

**Reporting summary**. Further information on research design is available in the Nature Research Reporting Summary linked to this article.

## Data availability

The genomic data used in this study are available in SoyBase [https://soybase.org/projects/SoyBase.C2021.04.php] and Ag Data Commons [https://data.nal.usda.gov/dataset/data-development-versatile-resource-1500-diverse-genomes-post-genomics-research]. The Williams 82 soybean reference genome sequence was downloaded from Phytozome v12 [https://phytozome.jgi.doe.gov/pz/portal.html]. The SoySNP50K iSelect Bead Chip data for reported soybean accessions in the US Soybean Collection were downloaded from SoyBase [https://soybase.org/snps/?msclkid=13bb233bc3d611ec9c6c9b130f971b5c]. The phenotypic data for reported accessions are available in Genetic Resources Information Network (GRIN) [https://npgsweb.ars-grin.gov/gringlobal/search]. Source data are provided with this paper.

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

## Acknowledgements

The authors would like to acknowledge Rick Meyer for his technical support in computational data processing and analysis, and Drs. Perry Cregan, Sam Wang, Zhixi Tian, Alice Kujur, David Goad, Ping Yates, and Guangqin Cai for reviewing the manuscript and providing valuable suggestions, Dr. Pengyin Chen for providing soybean seeds, Dr. Dilip Shah for his kind help with this project. The research used resource provided by the SCINet project and the AI Center of Excellence of the USDA-ARS. The research is funded by the United Soybean Board (USB Project #: 2020-152-0113, 2020-162-0202, 2220-152-0111), and jointly funded by United Soybean Board and Foundation for Food and Agriculture Research (USB-FFAR#: 2020-152-0118), and USDA-ARS (P`roject #: 5070-21000-042-00-D) to YQ.C.A.; D.H. was funded by China Scholarship Council.

## Author contributions

YQ.C.A. conceptualized the study. W.G., H.Z., and YQ.C.A designed and performed experiments, analyzed, and interpreted the data and wrote the manuscript; Y.L. performed expression assays; Z.Q. performed transgenic assays. H.J., D.H., Q.S., V.R.P., B.H.S., L.Y., and D.Y. contributed to data collection, data analysis and interpretation, and manuscript preparation and revision.

## Competing interests

The authors declare no competing interests.
