## [Peer Review File · Nature Communications]

POWR1 is a Key Domestication Gene Pleiotropically Regulating Seed Quality and Yield in SoybeanReviewers' Comments:

Reviewer #1:

Remarks to the Author:

This manuscript report identification and characterization of a locus associated with seed protein/oil content. Through GWAS, linkage mapping, and transformation analyses, the authors claimed this is a key domestication gene responsible for changes in seed protein/oil content. However, some of the materials and analyses were not described in details in the present manuscript. Additional analyses are needed to further validate the candidate gene and to understand how the gene function. Given that many polymorphisms between the wild soybeans and cultivated soybeans across the genome can be associated two distinct phenotypes of a domestication trait, it is important to show the distribution and relative contribution of all QTLs identified by GWAS and QTL mapping. To understand how the gene functions, profiling of gene expression should include both alleles and multiple tissues at multiple developmental stages. To validate the causal mutation(s), the transformation experiment should involve both alleles with the native promoters. The CDS of both alleles should be experimentally determined.

GWAS analyses using the wild and cultivated accessions often identify false positive association between a domestication-related trait (i.e., seed oil content in this study) and many clusters of SNPs polymorphic between such two subsets of accessions across the whole genome. It would be essential to articulate how the approach used in GWAS in this study avoided such false positives and if additional QTLs such as those presented in Zhou et al (Nature Biotechnology 2015, 33, 408-414; Fang et al., Genome Biology, 2017, 18, 161, and Zhang et al. 2018 Nature Plants, 4, 30-35) were also detected in other chromosomes/regions by GWAS. Theoretically, some of those QTLs are associated with SNPs/InDels distinguishable between the wild and cultivated soybeans. It would be clearer if the types (G. soja vs. G. max or landrace, elite variety) of the 278 accessions are mentioned in the main text, although those accessions are listed in the Table S1, and in Methods. Also, it should be articulate what types of the accessions mentioned in lines 70-77 and shown in Figure 1b and 1c and Figure 4c are. Based on the level of differences in oil/protein content and seed weight between compared accessions mentioned in line 74, it is very likely the authors were comparing the wild and cultivated accessions.

Variable seed colors particularly the dark colors affect the accuracy of the NIR measurement for seed oil content. How such effects were eliminated or reduced in linkage mapping? Did the phenotyping involve (multiple years/locations) replicates? Any additional major oil QTLs were identified by linkage mapping? What types of parental lines are those 15 mapping populations previously used for QTL mapping?

How many independent transformation events were obtained? Are expression of the transgene in the T2 plants from different events associated with effects on phenotypic changes? Why the Ubiquitin promoter rather than the native promoter of Glyma.20G085100 was used in the transformation experiment? Were transgenic lines grown in the field condition?

It is unclear how yield evaluation in the field condition was conducted – with or without multiple-year/location replicates.

How do the POWR-TE and POWR+TE function remains ambiguous. On one hand, transformation of POWR-TE into the POWR+TE background appear to increase protein content, which may suggest that POWR-TE is likely the wild-type “functional” allele; on the other hand, the authors pulled out gene expression data from a public dataset from Williams 82 carrying the POWR+TE allele. It is also important to compare the entire sequence of the two alleles including the regulatory region to pinpoint the casual mutations. Because the transformation experiment didn't use the promoters from either of the two alleles from the parental lines, the casual mutation for the phenotypic changes remains unclear. Comparison of the expression patterns of both alleles at different developmental stages

seeds, seed coat, and pods in parental lines as well as transgenic lines and the control would also help to interpret how the gene (allele) functions.

It is unclear what seed tissues at what stages used for profiling of gene expression in lines 167-191, and how a gene primarily expressed in seed coat affect the pathway genes involved in triacylglycerol (TAG) metabolism in other tissues. Profiling of the expression of the POWR-TE allele in those tissues appear to be critical for understanding how the allele function.

Line 193-199. Is PCA analysis limited to the POWR1 locus only? With so many accessions have been sequenced and re-sequenced (see Zhou et al., 2015; Feng et al., 2017; Shen et al. 2020 Cell, 182: 162). The presence and absence of the TE can be more precisely identified. How many of the 548 accessions were previously re-sequenced? Do the region surrounding the POWR locus show selective sweep?

Reviewer #2:

Remarks to the Author:

The authors present that a transposable-element inserted POWR1 has been selected for higher seed yield/weight/oil content and relatively lower protein during domestication for most cultivated soybean using the genetic analysis. However, the mechanism governing these important traits by a TE inserted in POWR1 is not clear. Due to the unknown function of POWR1, it may not be easy to address the mechanism. The authors are highly encouraged to exert more effort to interpret the RNA-seq data and discuss what they have found as RNA-seq are only data that may help to elucidate mechanisms in the manuscript. However, the interpretation of the RNA-seq data is too general. Some differentially expressed genes deserve further explanation. For example, How do the higher expression of CUT1 and SUC2 fit in POWR-TE?

It would be interesting to see whether the expression pattern of POWR1-TE from G. max is the same as POWR1+TE from G. max. The authors are also encouraged to discuss why Glycine max with POWR1-TE still produces higher seed yield/weight/oil content.

Fig. 2I: Although there is no difference in the overall expression of POWR1+TE and POWR1-TE, the variation is huge. Please provide the accession names for four and six G. max accessions carrying POWR1-TE and POWR1+TE in the methods, and compare the FPKM value of POWR1 for each accession.

Some figures lack a statistical description of error bars.

The seed coat expressed GmSWEET10a was selected for bigger seed size and higher seed oil during domestication. The authors missed this reference: Wang 2020 National Science Review. It is worth checking whether this gene is present in your DEG list?

REVIEWER COMMENTS

Reviewer #1 (Remarks to the Author):

1. This manuscript report identification and characterization of a locus associated with seed protein/oil content. Through GWAS, linkage mapping, and transformation analyses, the authors claimed this is a key domestication gene responsible for changes in seed protein/oil content. However, some of the materials and analyses were not described in details in the present manuscript.

Response: Thank you. In the revised manuscript, we added detailed information in the Materials and Methods section, including those for growth condition of NILs, seed measurement, association analysis method, soybean transformation, as well as germplasm types and details for new-added assays in the main text etc. e

2. Additional analyses are needed to further validate the candidate gene and to understand how the gene function.

Response: Thank you. We carried out additional assays to characterize the gene include (1) qRT-PCR to characterize two alleles' expression pattern in different tissues at different stages (2) to characterize expression for some oil and protein production related genes being affected by POWR1 in cotyledon and seed coat. (3) subcellular localization of two alleles in tobacco cells demonstrating that TE insertion alters its subcellular localization in nucleus (4) soybean transformation of POWR1 driven by the native promoter. All new results are helpful to elucidate gene function and were added to the revised manuscript.

I agree that there is a lot of experiments that we can do for in-depth study of molecular mechanism. However, the manuscript primarily reports the discovery of the gene and TE allele which require a great work to support the casual relationship and its function in soybean domestication and improvement and provides a broad insight into both its molecular mechanism. All the results presented here should provide good evidence to build a hypothesis, which regulating nutrient transporting in seed coat, for further in-depth study. We have worked on the genes for several years. The large amount of data that we presented in the manuscript should clearly provide a comprehensive picture of the POWR1 and TE in a broad perspective. We will pursue the in-depth study of the molecular mode of action and report findings in the next phase of our study. Unlike most model plants, soybean is one of most difficult species to transform, and also needs almost one year to get a positive transformed soybean, which is the major reason for us to take so long to finish the suggested experiments from reviewers. In addition, it often takes much more time to get soybean seed tissues than other tissues from model species with shorter life cycle and without going through almost the entire life cycle to get the tissues.

3. Given that many polymorphisms between the wild soybeans and cultivated soybeans across the genome can be associated two distinct phenotypes of a domestication trait, it is important to show the distribution and relative contribution of all QTLs identified by GWAS and QTL mapping.

Response: Great suggestion. We added the genome-wide QTLs that were identified using GWAS and QTL mapping in the revised manuscript (seed supplemental Figures 2,3). QTL mapping identified more than two major QTLs on chr15 and 20, of which chr15 QTL explains 16% for protein, 17% for oil, and chr20 QTL explains 23% of protein and 21% of oil variation. GWAS was carried out in a panel of diverse accessions and identified more than three significant loci on chr10,11, and chr20. It is very common that each RIL

mapping and GWAS analysis will generated multiple loci depending on the cutoff. Chr20 QTL identified in both linkage mapping and GWAS is the one being studied in the present study.

4. To understand how the gene functions, profiling of gene expression should include both alleles and multiple tissues at multiple developmental stages.

Response: This is a great suggestion. We immediately planted soybeans in the greenhouse and carried out the tissue specific expression for the two alleles using NILs for POWR1 using multiple tissues including root, leaf, stem, floral and developing seeds (seed coat and cotyledon) at early and middle maturation stages. The new result indicates that two alleles have similar spatial expression pattern in the selected tissues. In addition, genome sequence comparison of promoters of both alleles had no difference, also supporting no expression difference (see below). Following the tissue specificity assay, we examined DEGs from RNA-seq and some genes with known roles in protein or oil metabolism in different tissues including seed coat and cotyledon at the mid-maturation stage. All the newly-generated results provide comprehensive understanding of its molecular mechanism and we will continue the in-depth investigation of how it works in our near future study. Thank you.

5: To validate the causal mutation(s), the transformation experiment should involve both alleles with the native promoters. The CDS of both alleles should be experimentally determined.

Response: As mentioned above, soybean is among one of most difficult transforming species. It is also time-consuming. As you suggested, we immediately carried out transformation for the native promoter-driven POWR1-TE. It took us about 1 year to obtain the transformed seeds. The transgenic soybean expressing Ubi promoter and native promoter both reach a consistent result where protein was increased and oil was decreased, validating that POWR1 regulates the traits. In addition, our population genomics analyses and GWAS demonstrate the critical role of the TE indel in association with the seed traits, and our PCR-based assay confirmed the variant. To provide additional evidence supporting the causal mutation of TE, we also did subcellular localization of both alleles and demonstrate that TE insertion indeed altered POWR1 localization in nuclei.

6. GWAS analyses using the wild and cultivated accessions often identify false positive association between a domestication-related trait (i.e., seed oil content in this study) and many clusters of SNPs polymorphic between such two subsets of accessions across the whole genome. It would be essential to articulate how the approach used in GWAS in this study avoided such false positives and if additional QTLs such as those presented in Zhou et al (Nature Biotechnology 2015, 33, 408-414; Fang et al., Genome Biology, 2017, 18, 161, and Zhang et al. 2018 Nature Plants, 4, 30-35) were also detected in other chromosomes/regions by GWAS.

Response: Thank you for the insight about the issues, which many publications ignore it. During our research, we paid a great attention on it and used multiple approaches to validate our conclusion. GWAS is the first step to identify putative significant locus, and in the analysis we used MLM model to exclude confounding effects by population structure that is mainly caused by wild and cultivated soybean. The genome region containing POWR1 locus was identified in our study and Zhou et al (Nature Biotechnology 2015, 33, 408-414) and repeatedly reported in many other papers as indicated in our main

text (since 1992). The follow-up assays in our study confirm that POWR1 is the candidate gene, including linkage mapping using a RIL population, validating TE-phenotype correlation in RILs and previous RIL parental lines that have mapped POWR1 locus, and NILs for POWR1 locus, highly correlation between TE variation with protein and oil content in cultivated population, as well as the substantial increase in protein while decrease in oil content transgenic seeds (Ubi and native promoter). This series of experiments confirm that POWR1 is the gene for chr20 QTL. We added some information and cited these related papers in the revised manuscript. We also identified additional QTLs such as the ones in Chr11 and 15 and added them into the results.

7. Theoretically, some of those QTLs are associated with SNPs/InDels distinguishable between the wild and cultivated soybeans. It would be clearer if the types (G. soja vs. G. max or landrace, elite variety) of the 278 accessions are mentioned in the main text, although those accessions are listed in the Table S1, and in Methods.

Response: It is an excellent suggestion. We added the germplasm type (G. max or G. soja) in the needed places in the revised manuscript to make it clearer. Since most of the accessions were acquired from GRIN (Germplasm Resources Information Network) of USDA, and the accessions were mainly annotated with G. max or G. soja, not for landrace. To avoid confusion and minimize arbitrary error, we only indicated the germplasm type G. max or G. soja. Thank you for the great suggestion.

8. Also, it should be articulate what types of the accessions mentioned in lines 70-77 and shown in Figure 1b and 1c and Figure 4c are

Response: Thank you. We added the germplasm type information in these figures and other needed places to clearly indicated the germplasm type. In addition, we use Figure 1c to show the InDel visually in our sequence analysis, not intend to demonstrate that the correlation of the InDel with the seed traits. We revised our writing to clarify it.

9. Based on the level of differences in oil/protein content and seed weight between compared accessions mentioned in line 74, it is very likely the authors were comparing the wild and cultivated accessions.

Response: The panel used in Figure 1d is not the difference between G. soja and G. max, specifically, the comparison is made between 154 G. max-Ins (right column), and 116 G. soja-Del and 8 G. max-Del (left column). To avoid the confusion between G. max and G. soja, we split the POWR1-TE accessions into G. soja-Del, G. soja-Ins and compared their phenotypes. The comparisons indicate no or marginal differences between G. soja-Del and G. max-Del for the three seed traits suggesting that the Del has similar effects within G. soja or G. max. Meanwhile, both Del-carrying germplasms were dramatically lower in oil and seed weight and higher in protein than G. max-Ins accessions, suggesting that the observed phenotypic differences were primarily contributed by the Del rather than the overall difference between subpopulations. We added the new result and germplasm type in the revised manuscript. Thank you for the great comment.

10. Variable seed colors particularly the dark colors affect the accuracy of the NIR measurement for seed oil content. How such effects were eliminated or reduced in linkage mapping? Did the phenotyping involve (multiple years/locations) replicates? Any additional major oil QTLs were identified by linkage mapping?

Response: The equation for the NIR used in the experiment was developed based on the HPLC data from the University of Missouri Columbia analytical laboratory (Warrington et al., 2015). The effect was minimized by precise calibration using 900 North American soybean accessions and breeding lines; In addition, NIR measurement for seed oil has been used in many previous studies for seed oil and protein quantification and QTL identification; of the QTLs, Chr20 QTL was also repeatedly identified in various soybean populations using NIR measurements in multiple years and locations as we used in our study (Warrington et al., 2015; Patil et al., 2018; Nichols et al., 2006; Zhou et al., 2015; Bandillo et al., 2015). These studies indicate relative robustness of using NIR-measured oil and protein phenotype for major QTL detection in diverse soybean accessions. We added the detailed information in the revised manuscript.

In our study, linkage mapping with the RILs identified two major QTLs for oil and protein, one is the chr20 that is being presented in this study, the other major QTL is on chr15. We included the linkage mapping result in the supplemental figures of revised manuscript.

We added detailed information for MM section to the revised manuscript. Phenotypes in our study are from different sources and all were measured with the NIR machine. Phenotypes for the RILs were collected in two years (2012, 2015) and measured in two replicates; Phenotypes for the NIL plants were collected in two years in nine environments (Arkansas, Missouri, North Carolina, Knoxville, Springfield, Milan) and measured with the same NIR model (DA7250) as we did; Seed measurements for the natural population used in the present study were acquired from public database GRIN (<https://www.ars-grin.gov>). A previous study has demonstrated that there is no difference between for detecting the major oil and protein QTLs on chr20 by comparing adjusted and non-adjusted raw phenotypic data, which eliminated the concern of adverse effect of GRIN-sourced seed phenotypic data (multiple field locations across multiple years) on association power (Bandillo et al., 2015).

Warrington, C.V., Abdel-Haleem, H., Hyten, D.L., Cregan, P.B., Orf, J.H., Killam, A.S., Bajjalieh, N. et al. (2015) QTL for seed protein and amino acids in the Benning x Danbaekkong soybean population. Theor Appl Genet, 128, 839-850.

Bandillo NB, Lorenz AJ, Graef GL, Arquin D, Hyten DL, Nelson RL, et al. Genome-wide association mapping of qualitatively inherited traits in a germplasm collection. Plant Genome. 2017;10(2): WOS:000410819500002. pmid:28724068

Zhou, Z., Jiang, Y., Wang, Z., Gou, Z., Lyu, J., Li, W., Yu, Y. et al. (2015) Resequencing 302 wild and cultivated accessions identifies genes related to domestication and improvement in soybean. Nat Biotechnol, 33, 408-414.

Patil, G., Vuong, T.D., Kale, S., Valliyodan, B., Deshmukh, R., Zhu, C.S., Wu, X.L. et al. (2018) Dissecting genomic hotspots underlying seed protein, oil, and sucrose content in an interspecific mapping population of soybean using high-density linkage mapping. Plant Biotechnology Journal, 16, 1939-1953.

Nichols, D.M., Glover, K.D., Carlson, S.R., Specht, J.E., Diers, B.W. (2006) Fine mapping of a seed protein QTL on soybean linkage group I and its correlated effects on agronomic traits. Crop Sci, 46, 834-839.

11. What types of parental lines are those 15 mapping populations previously used for QTL mapping?

Response: We added the germplasm type (G. max, G. soja) in the main text and the figure and the TE variation in POWR1 the Supplemental Table S4. Out of the 7 pairs of high oil (+TE) and high protein (-TE) parents are G. max.

16. How many independent transformation events were obtained? Are expression of the transgene in the T2 plants from different events associated with effects on phenotypic changes? ? Why the Ubiquitin

promoter rather than the native promoter of Glyma.20G085100 was used in the transformation experiment? (?) Were transgenic lines grown in the field condition?

Response: We obtained two independent events. Quantitative real-time PCR using the POWR1-TE verified the increased expression of POWR1 in the POWR1+TE background, consistent with the phenotypic data of protein, oil and seed weight. In the two events, the event expressing higher POWR had more effect on the seed traits. In addition to the Ubi-promoter, as suggested, we carried out transformation for native promoter-driven POWR1-TE and the result validated the seed protein and oil. In the study, the transgenic plants were grown in the environment-controlled greenhouse that mimics field condition. We will try field test as long as we can have a permit. Since soybean transformation is not robust, successful rate varies a lot.

12. It is unclear how yield evaluation in the field condition was conducted – with or without multiple-year/location replicates.

Response: The NIL experiments in the field were carried out in two years in nine environments. We added the details information to the revised manuscript.

13. How do the POWR-TE and POWR+TE function remains ambiguous. On one hand, transformation of POWR-TE into the POWR+TE background appear to increase protein content, which may suggest that POWR-TE is likely the wild-type “functional” allele; on the other hand, the authors pulled out gene expression data from a public dataset from Williams 82 carrying the POWR+TE allele. It is also important to compare the entire sequence of the two alleles including the regulatory region to pinpoint the casual mutations. Because the transformation experiment didn’t use the promoters from either of the two alleles from the parental lines, the casual mutation for the phenotypic changes remains unclear. Comparison of the expression patterns of both alleles at different developmental stages seeds, seed coat, and pods in parental lines as well as transgenic lines and the control would also help to interpret how the gene (allele) functions.

Response: This is a great suggestion. As suggested, we investigated POWR1-TE and POWR1+TE in the different tissues in NIL lines for POWR1, including vegetative tissues root, leaf, stem, and reproductive tissues including flower, seed coat and cotyledon at the different stages. The qRT-PCR result indicates similar spatial expression patterns between POWR1-TE and POWR1+TE which is highly expressed in seed coat tissue at the early maturation stage.

As suggested, we compared 2 kb 5’ flanking region sequences of two RIL parents and found no sequence variants (Figure S5). It is also supported that we did not observe any DNA sequence variants in its flanking regions associated with protein, oil and weight in 265 accessions based on their whole genome re-sequences. This result is supported by the identical 2-kb promoter region between both alleles. We add the results to the revised manuscript. In addition, our subcellular localization indicated that TE insertion alters its localization pattern in nucleus.

14: It is unclear what seed tissues at what stages used for profiling of gene expression in lines 167-191, and how a gene primarily expressed in seed coat affect the pathway genes involved in triacylglycerol (TAG) metabolism in other tissues. Profiling of the expression of the POWR-TE allele in those tissues appear to be critical for understanding how the allele function.

Response: The RNA-seq was performed using the whole developing seeds at the mid-maturation stage. Some genes associated with TAG metabolic pathway were identified, suggesting that those genes in TAG pathways in seeds might be affected by POWR1. We further investigated expression of several selected genes in the TAG pathway (such as DGAT1, AAE, GPAT et al) in developing seeds and the result validated the RNA-Seq results. On the other hand, we compared POWR1+TE and POWR1-TE in different tissues including root, leaves, flowers, stem, developing whole seeds and seed coat and cotyledon at the early and middle seed-maturations stages. Two POWR1 alleles have similar spatial expression patterns. Regarding expression of powr1 in seed coat affect the expression of other genes in the other tissues, one of the speculations could be that altering function of POWR1 lead to physiological changes such as sucrose transporting to cotyledon, and then the altered sucrose level induces or suppresses transcriptional expression of TAG in cotyledon and other tissue. Some of the changes are secondary and sequential transcriptional response to POWR1 variations.

15. Line 193-199. Is PCA analysis limited to the POWR1 locus only? With so many accessions have been sequenced and re-sequenced (see Zhou et al., 2015; Feng et al., 2017; Shen et al. 2020 Cell, 182: 162). The presence and absence of the TE can be more precisely identified. How many of the 548 accessions were previously re-sequenced? Do the region surrounding the POWR locus show selective sweep?

Response:

- a. *The PCA was made using genome wide SNPs.*
- b. *The genomes for all the 548 accessions used in our study were all re-sequenced and used in the study. We revised the manuscript to make it clear. The analysis also included whole genome sequences of 91 soja accessions representing diversity of soja in US collection, which we sequenced. The whole genome sequences of 548 diverse accessions played a significant role in the study. The study should be a good demonstration project of post-genomics or translational genomics on how to apply the large number of whole genome sequencing in biological discovery and study.*
- c. *We used genome sequence of the 4 Mb region for regional "GWAS" at single nucleotide resolution, which provide much stronger evidence for the gene and causative allele. Investigating TE variation in newly-published data is a great suggestion, we downloaded and re-analyzed sequencing data from recent studies. In total, sequencing data for 5,226 genomes were reanalyzed, of which, 3,956 accessions (3,451 G. max, 363 G. soja) carrying clear genotyping on TE locus and thus was used for TE genotyping analysis. This large-scale analysis indicated that 3,451 (96.05%) of G. max accessions carry POWR+TE, whereas, 361 (99.45%) of G. soja accessions carry POWR1-TE. The TE genotyping result in the large-population is fully consistent to our current result (94.7% G. max carry POWR+TE, 99.3% G. soja carry POWR-TE), fully supporting the result of 548 accessions.*
- d. *Yes. POWR1 was located within an approximately 520-kb sweep region (chr20: 31,641,057 - 32,160,913) as inferred by Tajima's D of < -2 (Fig. 3c) and high G. soja/G. max π ln-ratios (larger than 2.4) (chr20: 31,654,290 - 32,157,761).*
- e. *In addition, since we analyzed whole genome sequences of the 548 accessions, we did an exhausted examination of the genes and DNA variants in the QTL region, showing that POWR1 and TE is the causative gene and allele. The whole genome sequences also allow us to identify post-domestication POWR1 allele transfer, and to demonstrate 100% correlation of the TE alleles with soja and max after removing the accessions with the allele transfer, which shows that TE insertion is a key event in soybean domestication. The results from multidisciplinary approaches should provide multiple cross-validation for the functions of POWR1 and TE and serve much stronger evidence than from a single approach. With the newly added data, the manuscript*

should be much more convincing and insightful. Thank you for your time to review the manuscript and provide many valuable suggestions.

Reviewer #2 (Remarks to the Author):

1. The authors present that a transposable-element inserted POWR1 has been selected for higher seed yield/weight/oil content and relatively lower protein during domestication for most cultivated soybean using the genetic analysis. However, the mechanism governing these important traits by a TE inserted in POWR1 is not clear. Due to the unknown function of POWR1, it may not be easy to address the mechanism.

Response: We appreciate this reviewer's time spent on reviewing our manuscript and provide great suggestions. With the suggestion, we arranged additional experiments, including investigating two alleles' expression in different tissues and developing seeds at different stages, comparing expression for selected genes under the POWR1+TE or POWR1-TE background, comparing POWR+TE, POWR-TE in subcellular localization, soybean transformation assay with the native promoter. We added the result to the revised manuscript and the result increased our understanding how POWR1 functions.

2. The authors are highly encouraged to exert more effort to interpret the RNA-seq data and discuss what they have found as RNA-seq are only data that may help to elucidate mechanisms in the manuscript. However, the interpretation of the RNA-seq data is too general. Some differentially expressed genes deserve further explanation. For example, How do the higher expression of CUT1 and SUC2 fit in POWR-TE?

Response: To increase the understanding for the RNA-Seq data, we carried out gene expression examination for the genes such as SUC2, OLEO, WRI1, DGAT1 that putatively affected by POWR1 using qRT-PCR in near isogenetic lines for POWR1. The result provide additional evidence for us to develop the proposed model in the main text and we added it to the revised manuscript. In addition, subcellular localization and gene expression analysis shows that TE insertion changed exclusive localization of POWR1-TE in nucleus, not gene expression in our effort to understand its molecular effort.

3. It would be interesting to see whether the expression pattern of POWR1-TE from G. max is the same as POWR1+TE from G. max.

Response: It is a good suggestion. We examined the expression for both alleles in soybean and two alleles exhibited similar expression pattern in tissues. We added the new result to the revised manuscript. Thank you.

4. The authors are also encouraged to discuss why Glycine max with POWR1-TE still produces higher seed yield/weight/oil content.

Response: Thank you. We added some discussion in the revised manuscript.

Soybeans seed yield/weight and oil/protein are complex traits controlled with many genes, some with small effects, and thus far as we known, there are over 300 hundred loci in soybean genome affected oil and protein content. Therefore, the measured seed traits such as oil and protein are accumulative effects of all major and minor QTLs, POWR1 locus is a major one of them and other loci may also contribute to the trait variation such as our recently identified SWEET39 for Chr15 QTL that we also work on. It is likely

that many *G. max* with POWR1-TE carries high oil/weight alleles on the other loci, leading to high oil/seed weight relative to wild soybean. Nevertheless, it is important to uncover POWR1 locus with the major effects on seed protein, oil and seed weights to further understand the underlying molecular and genetic mechanism.

5. Fig. 2I: Although there is no difference in the overall expression of POWR1+TE and POWR1-TE, the variation is huge. Please provide the accession names for four and six *G. max* accessions carrying POWR1-TE and POWR1+TE in the methods, and compare the FPKM value of POWR1 for each accession.

Response: We added the 10 accessions and the POWR1 expression level to a supplemental Table S6. All of the 10 accessions are G. max, which were selected to avoid the potential variation between G. soja and G. max. In addition, we added RT-PCR data to confirm that the two alleles have similar expression patterns. T.test indicate two groups do not have significant difference in POWR1 ($p=0.12$). Average FPKM for POWR1+TE is 0.67 ± 0.19 and POWR1-TE is 0.97 ± 0.29 .

6. Some figures lack a statistical description of error bars.

Response: Thank you. We added needed description as suggested. It is a standard derivation (SD)

7. The seed coat expressed GmSWEET10a was selected for bigger seed size and higher seed oil during domestication. The authors missed this reference: Wang 2020 National Science Review. It is worth checking whether this gene is present in your DEG list?

Response: Thank you. The SWEET was not in the DEG list and we further validated it in qRT-PCR analysis. Reference was added.

Just a note, the study devotes most of the effort to providing evidence proving that POWR1 and TE insertion are the causes for the QTL for the important traits and it is a key domestication gene. There are many experiments that we can do to gain insight into molecular mode of action for the important and exciting gene. Hope that the results presented on the gene, mutation and mode of action, and hypothesis provide a solid starting point for future investigation. In addition, unlike model plant species, it takes us one year to generate the transgenic plants containing its native promoter driven POWR1-TE to validate its effect on seed traits, which it is the reason for us to get the revised manuscript back to you so late. We already have worked on the gene for several years and generated a good number of significant discoveries in a broad perspective. We are characterizing molecular mode of action in more and will report in a separate following-up paper. Thank you again for your time and kindness to review the manuscript and provide many constructive comments.

Reviewers' Comments:

Reviewer #1:

Remarks to the Author:

Most of my concerns have been addressed or reasonably argued, and I do not have additional comments.

Reviewer #2:

Remarks to the Author:

Thanks to the authors for addressing my concerns.

Line 157, Fig. 2h statically showed that POWR1+TE and POWR1-TE are expressed statistically differently, especially in seed coat and embryo. The authors seem to ignore the differences they presented. It is not convincing to state "suggesting TE insertion unlikely affect their expression pattern." (line 157 and 158) and "Thus, both gene expression and sequence comparisons suggest that TE insertion cause variation of seed traits likely through altering protein activity, not gene expression." (line 162 and 164).

For the expression data, it is clear TE insertion likely affects gene expression. The authors may want to investigate longer promoter sequences, although they did not observe an evident sequence variation when comparing the 2-kb upstream sequences (including 5'utr) of the start codon of POWR1+TE with that of POWR1-TE.

Figs. 2h and 2j missed statistical information. The numbers of stars on SUC2 and SUC4 may be messed up. Please confirm.

Fig. 2g, It seems no localization difference between GFP-POWR1+TE and a free GFP. This should be better done by a co-localization experiment.

Based on this observation, I would ask how the authors know POWR1+TE is a functional protein since there is no evidence to support its direct function in the current manuscript.

Fig 3a, It is interesting to see UbiOE-2 has much higher protein but not much lower oil content compared to WT. Do you have any possible explanation for this?

Line 328: SWEET genes were initially characterized and named based on their phylogenetic relationships. In your letter, the SWEET39 you work on was not named using this system, which confused future readers.

Point-By-Point Response to The Reviewers' Comments

Reviewer #1 (Remarks to the Author):

Most of my concerns have been addressed or reasonably argued, and I do not have additional comments.

Response: Thank you for the time to review the MS.

Reviewer #2 (Remarks to the Author):

Thanks to the authors for addressing my concerns. Line 157, Fig. 2h statically showed that POWR1+TE and POWR1-TE are expressed statistically differently, especially in seed coat and embryo. The authors seem to ignore the differences they presented. It is not convincing to state "suggesting TE insertion unlikely affect their expression pattern. " (line 157 and 158) and "Thus, both gene expression and sequence comparisons suggest that TE insertion cause variation of seed traits likely through altering protein activity, not gene expression. " (line 162 and 164). For the expression data, it is clear TE insertion likely affects gene expression. The authors may want to investigate longer promoter sequences, although they did not observe an evident sequence variation when comparing the 2-kb upstream sequences (including 5'utr) of the start codon of POWR1+TE with that of POWR1-TE.

Response: Thank you for the careful review and point out the inconsistency. We revised to focus on describing the data and their expression in flower and seed coat while point out the difference in the other tissues. I feel that it may not have biological significance due to low level expression in those tissues.

Figs. 2h and 2J missed statistical information. The numbers of stars on SUC2 and SUC4 may be messed up. Please confirm.

Response: Thank for pointing it out. We added the statical information in figure legends. We update the statistical information for SUC2 (top panel) which is significant level at a level of $p < 0.001$.

Fig. 2g, It seems no localization difference between GFP-POWR1+TE and a free GFP. This should be better done by a co-localization experiment.

Based on this observation, I would ask how the authors know POWR1+TE is a functional protein since there is no evidence to support its direct function in the current manuscript.

Response: We do not intend to state that POWR1+TE is a functional gene. The evidence presented in the MS cannot prove that POWR1+TE is a functional protein or not. However, the difference in localization of GFP-POWR1+TE and -TE in nucleus indicated that the TE insertion

alters the localization of the POWRI. It could be from loss-of-function or altered function of POWRI. We revise MS to emphasize it.

Fig 3a, It is interesting to see UbiOE-2 has much higher protein but not much lower oil content compared to WT. Do you have any possible explanation for this?

Response: *Generally, 1% increase in oil correlated to 2% decrease of protein in soybean, therefore, we generally see less change in oil than protein in UbiOE-2.*

Line 328: SWEET genes were initially characterized and named based on their phylogenetic relationships. In your letter, the SWEET39 you work on was not named using this system, which confused future readers.

Response: *Agree. The systems for naming SWEET in current published papers are confusing and not consistent. We used the gene name assigned by the publication by Patil (BMC Genomics, 2015 and another related paper on SWEET39 (Miao et al 2020), which both studies focus on soybean SWEET genes. To connect these highly related studies, we decided to use SWEET39 in this study and to avoid confusion. We also corrected the typo “GmSWEET” to “GmSWEET39” in the revised manuscript.*

Patil G, Valliyodan B, Deshmukh R, Prince S, Nicander B, Zhao M, et al. Soybean (Glycine max) SWEET gene family: insights through comparative genomics, transcriptome profiling and whole genome re-sequence analysis. BMC Genomics. 2015; 16:520. <https://doi.org/10.1186/s12864-015-1730-y> PMID: 26162601; PubMed Central PMCID: PMC4499210.

Miao L, Yang S, Zhang K, He J, Wu C, Ren Y, et al. Natural variation and selection in GmSWEET39 affect soybean seed oil content. New Phytol. 2020; 225(4):1651–66. <https://doi.org/10.1111/nph.16250> PMID: 31596499.

Reviewers' Comments:

Reviewer #2:

Remarks to the Author:

I was not recommending the authors use SWEET39 in this manuscript or future publications. Those two papers listed by the authors did not follow the original name system, which messed up SWEET names in soybean and confused readers.

How to name SWEETs should follow these three references below to benefit the whole community.

Eom, J.-S., Chen, L.-Q., Sosso, D., Julius, B.T., Lin, I., Qu, X.-Q., Braun, D.M., and Frommer, W.B. (2015). SWEETs, transporters for intracellular and intercellular sugar translocation. *Curr. Opin. Plant Biol.* 25: 53–62.

Wang, S. et al. (2020). Simultaneous changes in seed size, oil content and protein content driven by selection of SWEET homologues during soybean domestication. *Natl. Sci. Rev.* 7: 1776–1786.

Wang, S., Yokosho, K., Guo, R., Whelan, J., Ruan, Y.-L., Ma, J.F., and Shou, H. (2019). The soybean sugar transporter GmSWEET15 mediates sucrose export from endosperm to early embryo. *Plant Physiol.* 180: 2133–2141.

Response to the Reviewer's Comments

Response to Referees Letter of the Third Review

Reviewer #2 (Remarks to the Author)

I was not recommending the authors use SWEET39 in this manuscript or future publications. Those two papers listed by the authors did not follow the original name system, which messed up SWEET names in soybean and confused readers.

How to name SWEETs should follow these three references below to benefit the whole community.

Eom, J.-S., Chen, L.-Q., Sosso, D., Julius, B.T., Lin, I., Qu, X.-Q., Braun, D.M., and Frommer, W.B. (2015). SWEETs, transporters for intracellular and intercellular sugar translocation. *Curr. Opin. Plant Biol.* 25: 53–62.

Wang, S. et al. (2020). Simultaneous changes in seed size, oil content and protein content driven by selection of SWEET homologues during soybean domestication. *Natl. Sci. Rev.* 7: 1776–1786.

Wang, S., Yokosho, K., Guo, R., Whelan, J., Ruan, Y.-L., Ma, J.F., and Shou, H. (2019). The soybean sugar transporter GmSWEET15 mediates sucrose export from endosperm to early embryo. *Plant Physiol.* 180: 2133–2141.

Response: The name was changed to SWEET10a. We changed the statement to “SWEET10a gene (also named as GmSWEET39)”, which both names are referred. Thank you for your time and providing the constructing comments over the reviewing process.